# Social Connections and Self-Perceived Depression: An Enhanced Model for Studying Teenagers’ Mental Wellbeing

**DOI:** 10.3390/ijerph192315791

**Published:** 2022-11-27

**Authors:** Abdullah Sarwar, Md. Amirul Islam, Muhammad Mohiuddin, Mohammad Ali Tareq, Aysa Siddika

**Affiliations:** 1Faculty of Management, Multimedia University, Cyberjaya 63100, Malaysia; 2Faculty of Business Administration, Laval University, Quebec City, QC G1V0A6, Canada; 3Department of Finance, Faculty of Business and Economics, University Malaya, Kuala Lumpur 50603, Malaysia

**Keywords:** self-perceived depression, social connection, mental wellbeing, teenagers, Malaysia

## Abstract

The rising prevalence of depression among teenagers in Malaysia as well as globally makes it a vital issue to study. The purpose of this research is to examine the effects of social connection and self-perceived depression towards the improved mental wellbeing of the teenagers of Malaysia. Moreover, the mediating role of self-perceived depression on the improvement of the mental wellbeing of teenagers is examined in this study. This study followed a questionnaire-based approach. The sample of this study included 289 students aged between 15 and 19 years from Klang Valley, Malaysia. Prior permission was obtained from school authorities as well as from parents to allow their children to participate in the survey. To find out the structural relationship between the variables, PLS-SEM was utilized. This study finds that stronger social connections with family and friends may result in reduced self-perceived depression among Malaysian teenagers. Moreover, self-perceived depression among the teenagers surveyed had a negative effect on their improved mental wellbeing. The findings of this study will significantly affect how depression theories are currently understood and have consequences for social work, services, and policy interventions regarding teenagers in Malaysia.

## 1. Introduction

Adolescent depression is a widespread issue in many countries [1]. School-aged children often have poor mental health since they frequently display indicators of sadness and hopelessness [2]. According to World Health Organization reports, globally 14.3% of 10–19 year-olds experience depression, anxiety, and behavioral disorders due to poor mental wellbeing [3]. Interpersonal and social interactions are frequently recognized as the most significant predictors of teenager psychosocial wellness despite the fact that analysts have linked several interconnected aspects to teenagers’ sadness [4]. Teenagers who have healthy social connections with their family and friends are better able to manage stress, and their mental wellbeing may be predicted [5].

As a result of globalization and the increased use of communication technology, younger generations are growing up with smartphones and spend far less time interacting face to face with family and friends [6,7]. Teenagers’ growing reliance on the Internet and social media tends to lead to fewer affectionate social interactions, less time spent with their family and friends, a deterioration in their ability to focus and study in class, and increased exposure to online games, violence, and bullying, all of which are detrimental to their mental health [8,9].

Teenage schoolchildren often have poor mental health since they frequently display indications of sadness and hopelessness [10,11,12]. Personal and family problems and the influence of social media are mentioned as the contributors to their stress. According to the National Health and Morbidity Survey [NHMS] of 2019, depression is the second most frequent mental health concern among Malaysian teenagers, who account for around 29.9% of the country’s youth, or 4.2 million people [13]. A total of 7.9% of children between the ages of 5 and 15 were found to have mental health issues; this figure had increased from results recorded in 2015 mostly because of poor peer interaction [12]. Moreover, in the NHMS 2017 Adolescent Health and Nutrition Survey, the key findings reported that 20% of Malaysian children aged 13–17 years are depressed, 40% of them are anxious, and 10% are in stress [13]. This suggests that teenage depression is a significant public health concern that requires prompt attention. 

Analysts have identified a number of interrelated elements that contribute to teenage depression. Self-esteem, interpersonal stressor sources, a low level of self-efficacy, and a high level of perceived stress are few to mention [14]. In addition, Bruine et al. acknowledged the link between larger social networks and higher levels of mental wellbeing across the life span [15]. By examining the relationship between parental engagement and close friendships as indicators of social support networks and depression among teenagers in South Asia, Mursid [16] found that social support is a social predictor of adolescent mental health [17]. Children who have more social support may have less anxiety, behavioral issues, and depressive symptoms [18]. Less social support, on the other hand, has a detrimental impact on resilience levels, which are crucial for preventing depressive symptoms. Moreover, the guidelines for mental health promoting preventative interventions for adolescents by WHO put emphasis on research on the impact of involving parents, caregivers, and families in psychological interventions [19].

Therefore, the purpose of this study is to investigate how social connections (such as family and friendship) among teenagers impact how depressed they perceive themselves to be and how these interactions contribute to their mental wellbeing.

## 2. Literature Review

Depression is typically brought on by one or more stressful, depressing, or disappointing life events, which are then exacerbated by issues with family or friends, persistent issues in education, and specific personality features [20]. Teenager self-reported low mood was shown to have a cohort impact in research [21], pointing to systemic factors. However, research implies that those who have strong social support networks are better at handling stressful life events and getting over psychiatric issues [18]. Hence, this study will utilize Ryff’s [22] psychological wellbeing theory as the foundation for the proposed model which will be linked with several other social connection-related variables that lead to teenagers’ self-perceived depression and mental wellbeing.

Wellbeing is perceived in various ways in different fields of study. In the psychology field, the majority of researchers agree that wellbeing illustrates optimal psychological experience and functioning in life [23]. According to Ryff [24], wellbeing is currently referred to by three paradigms that include psychological wellbeing, subjective wellbeing, and composite wellbeing. The extensive theoretical literature has highlighted the function of psychological wellbeing [16,18,22]. In the present study, psychological wellbeing is also used to represent mental wellbeing. In this study, Rogers’ idea of the fully functional person is combined with Maslow’s idea of self-actualization, Allport’s idea of maturity, and Jung’s notion of individuation [25]. 

### 2.1. Mental Wellbeing

Every person must be in a condition of good mental health to fulfill their full potential, be able to deal with everyday challenges, be productive and successful at work, and be able to give back to their community [26]. The wellbeing and best possible development of a child in their emotional, behavioral, social, and cognitive domains are collectively known as child mental health [27]. Due to the distinctive developmental milestones that children go through, child mental health is frequently described as being distinct from adult mental health and as having additional facets. The wellbeing of a child is significantly influenced by factors such as the child’s gender, heredity, family, community, and larger society [28,29]. 

Teenagers’ psychological wellbeing encourages good feelings, allowing them to experience contentment and enjoyment in their lives. For future development, the positive experience of mental wellbeing is highly essential [30] Psychiatric wellbeing in teenagers, according to Basson [30], is highly significant as a protective factor in order to decrease negative impacts on adolescents, such as depressive disorders, anxiety, loneliness, and juvenile delinquency, such as underage drinking, smoking, and drug usage. 

Additionally, studies found that teenagers who are in good mental health may have a more favorable experience in their intrapersonal and interpersonal relationships, in their academic life, and future development [31,32]. A lack of psychological wellbeing in the academic sector [33] is the root cause of teenagers getting into fights and skipping class.

A good quality of life is dependent on one’s level of wellbeing [34]. The state of being in which an individual has positive attitudes toward themself and others, has the ability to make decisions and regulate their own behavior, to regulate their external environment, has a sense of purpose in life, and has the capacity for personal growth is known as psychological wellbeing [20]. According to Diener and Suh [35], the fundamental components of psychological wellbeing are contentment with one’s life and positive affect. When we talk about affect, we refer to both positive and negative feelings and emotions; however, when we talk about life satisfaction, we refer to psychological pleasure with one’s life. There are six aspects of psychological wellbeing experienced by people according to Ryff et al. [36]; these dimensions include self-acceptance, having good relationships with others, autonomy and environmental mastery, having a sense of meaning in life, and personal progress. 

### 2.2. Young People and Depression

Numerous studies have shown that a sizable proportion of school-aged people in the general population experience depression. According to recent US data, the proportion of adolescents aged 12 to 17 who reported having serious depression symptoms increased from 8.7 to 13.2% between 2005 and 2017 [20]. 

Among children and adolescents worldwide, 10% to 20% have a mental health condition, according to global estimates [37]. Differing geographic areas have different rates of depression than others because of a variety of reasons, including industrialization, socioeconomic levels, and the availability of support services in each place. Depressed teenagers are at greater risk of developing psychosocial impairments and being admitted to hospital more often. Furthermore, depression might have a negative impact on the academic achievement of these teenagers as well as on their socializing [17,38].

Because the transition from childhood to adolescence is accompanied by significant mental and bodily changes, the appearance of psychosocial issues, such as depression, is typical throughout this period of one’s development. Adolescent depression interferes with the ability to function in daily life and activities [39]. Adolescents suffering from depression are more likely to engage in antisocial conduct and to misuse alcohol and other substances [38,39]. These issues justify the importance of examining the mental health of teenagers in Malaysia. 

Malaysian teenagers account for one fifth of the country’s population, and more than ten percent of them are depressed, according to official statistics [40]. A further point to mention is that 5.2% of secondary school teenagers in Malaysia are suffering from serious depressive symptoms [41]. The ethnic, economic, and cultural origins of individuals in a place may produce anxiety and worry among the general public [42,43].

In most teenagers, depression is difficult to diagnose because it is seen as a typical transitional difficulty that must be dealt with rather than as a mental health issue to be treated. Teenage depression is often associated with a variety of life challenges, a lack of family relationships, a lack of support from family members while coping with life stresses, disagreements in the family, and financial difficulties. Furthermore, poor peer relationships, such as having no friends, failing to communicate with friends, and feeling unaccepted by friends, may all contribute to adolescent melancholy. In certain situations, depression may lead to incidences of suicide and other risky activities among youths [21]. 

### 2.3. Social Connection

Recent research by Liu and Merritt [44] suggested that there were often mild to moderate non-trivial connections between parenting and childhood depression. According to research, depression and parental communication styles are directly related. Numerous studies have shown a significant link between depression and family contact with parents and other family members [45]. In an effort to understand how changes in family cohesion during the high school to college transition may be related to changes in depressive symptoms, Joao et al. [46] discovered that students who reported increases in family cohesion reported declines in depressive symptoms during the college transition. Social support derived from social connection includes different forms, such as instrumental, emotional, and informational resources and social companionship. Forms of social support can be further classified into received social support and perceived social support. Being subjective in nature, perceived social support has proved to be more significant to the effects of mental wellbeing on a person [47]. 

Therefore, the present research predicts a significant inverse relationship between social support and connections and teenagers’ self-perceived depression. Joao et al. [46] found that students who reported improvements in family cohesiveness showed a decrease in depressive symptoms over the high school to college transition, which helped the researchers better understand how changes in family cohesion may be associated with changes in depressive symptoms. In another study, Badri et al. associated the significance of social connection with self-perceived depression among adolescents [48]. Likewise, the current study indicates a substantial association between social contacts and support and self-perceived depression among teenagers.

Social interaction, one of the most important and widely researched factors affecting mental health, includes social support, social networks, social bonds, and social capital [49,50,51]. Social support may be defined as the aid provided by members of a person’s social network who fulfill the role of their ego in that network. Teenager depression in relation to relationships with parents is perceived through various characteristics, and research outcomes are often contradictory and multifaceted [52]. Moreover, in a national survey of mental health and wellbeing among 8841 Australians, social support and interactions were studied at the level of depression [53]. 

The most important predictors of teenage psychosocial health are typically thought to be social and interpersonal relationships [9]. Positive relationships with family and friends are one element that may help children cope with stress and be used to predict their mental health. One of the matters that teenagers clearly perceive as a central theme during their teenage period is that of peer-to-peer relationships and allocation among peers, as the amount of time spent with their friends in school, during extracurricular leisure, and during sports activities as well as the time devoted or spent participating in hobby activities with their peers is regarded as being more significant than the time spent with their family [52].

## 3. Research Framework and Hypotheses

Figure 1 offers a suggested study framework that is based on the theoretical and empirical literature that was previously discussed. It demonstrates how teenagers in Malaysian family and friendship groups influence their self-perceived despair and how interactions improve their mental wellness. The following possibilities are offered:

**H1.** 
*There is a negative relationship between social connection and self-perceived depression.*


**H2.** 
*There is a negative relationship between self-perceived depression and improved mental wellbeing.*


**H3.** 
*There is a positive relationship between social connection and improved mental wellbeing.*


**H4.** 
*There is a mediating effect of self-perceived depression between social connection and improved mental wellbeing.*


## 4. Methods

This study followed a questionnaire-based approach that was distributed randomly among Muslim teenagers (aged 15–19) studying at various public and private schools and colleges within Peninsular Malaysia. Prior permission was obtained from school and college authorities as well as from parents to allow their children to participate in the survey. 

The gathering of data was the most important aspect of this study. To conduct the study, various private and public schools were randomly chosen. The researcher initially spoke with the administrators of the chosen schools to inform them of the current study project. Later, the school administration was asked to give the questionnaire and consent letter to the teenage students and instruct them to bring it home so that their parents could sign the consent form and authorize their participation in the survey. The goal of the research and how their privacy would be protected were both spelled out in the permission letter in full. The researcher was notified by the relevant school counsellors of the number of returned consent forms after a few weeks. To guarantee participant confidentiality, all surveys were coded and free of any personal information.

The measurement scales of three constructs were adopted from the existing literature [9,21,52]. Social connection was measured with a two-item scale, namely the amount of quality time spent with family and frequent meeting with friends and relatives. Self-perceived depression and improved mental wellbeing were measured using a three-item scale and six-item scale, respectively. The measurement scale for self-perceived depression was a three-item scale, namely how often they felt downhearted and depressed, how often they felt isolated from the people around them, and their feelings about the trustworthiness of people. The improved mental wellbeing scale included involvement in sports group, in heritage groups, religious or spiritual groups, informal activities with friends, and the respondent’s personal feelings about their overall mental wellbeing and satisfaction with family life. All scales were measured by five-point Likert agreement scale. 

This study used random sampling to ensure the largest possible sample. Prior to the final study, a pilot study was conducted among 30 teenage students. The questionnaires were distributed among 350 students and the final sample size for this cross-sectional study reached 289. 

The data were analysed using PLS-SEM to test and validate the proposed framework. Partial least squares structural equation modelling (PLS-SEM) is a method of structural equation modelling that enables the estimation of cause–effect relationships with latent and observed variables. It also enables the analysis of relationships of multiple constructs, indicator variables, and structural paths without any distributional assumptions on the data [54]. Being exploratory in nature, the current study uses PLS-SEM analysis because it enables simultaneous examination of the constructs and underlying structural model. Moreover, PLS-SEM is able to analyze the formative and reflective models along with potential advantages over linear regression model. Therefore, it is the preferred way to evaluate path diagrams for latent variables and multiple indicators [54].

Convergence validity, discriminant validity, and indicator reliability were utilized to assess the measurement model’s internal consistency reliability [55,56]. Additionally, path co-efficient evaluation, coefficient of determination, and predictive relevance assessment were utilized to examine the structural model’s multicollinearity [57]. 

## 5. Results 

Table 1 presents the demographic profile of the respondents. Among the respondents, 59.9% were male and 40.1% were female. In the present study, the majority of the respondents (70% or 202 students) were teenagers in the group of 15–19 years of age. A total of 40% (n = 87) of the teenagers were below 15 years of age. According to the educational stage, 36% (n = 105) of the respondents were studying at a higher school level and 59.8% (n = 182) of the respondents were at the bachelor level. The rest of the respondents (4.2%) were at the diploma level. 

To ensure the validity and reliability of the model, it is imperative to assess the measurement model. In this study, the Cronbach’s alpha values for the three constructs are 0.90, 0.83, and 0.64, and the composite reliability ranges from 0.848 to 0.926, which is higher than the threshold level of ≥0.70 required to be reliable [58]. The accepted level for average variance extracted (AVE) is ≥0.50. In the present study, the results showed that the AVE ranged from 0.68 to 0.75, which is far above the standard value that validates the constructs. Therefore, this model achieved construct reliability and validity presented at Table 2. 

Moreover, through the heterotrait–monotrait ratio (HTMT) [59], the discriminant validity was tested. The HTMT value should be ≤0.90 to be deemed significant. From the results (Table 3), it was observed that all of the construct’s values for HTMT were less than 0.90, which confirms that the constructs were different and not related to each other. Consequently, the model ascertains the discriminant validity. 

### 5.1. Structural Model Assessment 

In assessing the structural model, the first step is to measure the predictive power of the model through R^2^, Q^2^, and f^2^. The standard values required for R^2^, Q^2^, and f^2^ are ≥0.02, >0, and ≥0.02, respectively. In this study, R^2^, Q^2^, and f^2^ meet the required levels (Table 4 and Table 5). Although the coefficients of the determination (R^2^) value for the constructs are weak, the Q^2^ values assure the predictive relevance of the path model. Moreover, the f^2^ values for improved mental wellbeing interpret medium and self-perceived depression as having low relevance to the constructs in explaining the selected endogenous constructs. 

### 5.2. Hypothesis Testing 

To test the relationship between the variables, hypothesis testing was conducted. The structural model is presented in Figure 2, and Table 6 shows the results of the model. A one-tailed analysis was conducted. The minimum t statistics required to be significant are ≥1.96 [60].

H1, H2, and H3 express the direct relationship between the constructs, and all of them are supported. Self-perceived depression (t statistics = 6.691, *p* value = 0.00) and social connection (t statistics = 5.13, *p* value = 0.00) were found to have significant effects towards improved mental wellbeing. Similarly, social connection is significantly and inversely connected with self-perceived depression.

For the mediation effect, this study finds that self-perceived depression mediates the relationship between social connection and improved mental wellbeing (t = 3.606, *p* value = 0.00).

## 6. Discussion

This study showed that social connection had a significant inverse effect on self-perceived depression. This result is consistent with the findings of Badri et al. [9], Nieuwsma et al. [43], and Subramanyam et al. [45]. It implies that an increased level of social connection negatively affects self-perceived depression among teenagers; therefore, it denotes that the critical predictor of the psychosocial wellbeing of teenagers is the improvement of social ties or connectivity that decreases depressive symptoms among teenagers during the transition period of their lives. Thus, encouraging teenagers to engage in group activities, such as sports and travel as well as other socio-cultural activities, can be helpful for their mental development.

Similarly, self-perceived depression has a negative effect on the improved mental wellbeing of teenagers. Their frequently high levels of feelings of depression and isolation from and distrust of the people around them make teenagers depressed and inhibit their mental wellbeing. Studies by Deng et al., Badri et al., and Yu et al. revealed similar effects of self-perceived depression on mental wellbeing. Authors [47,61,62] and the IPH’s [13] report on Malaysian adolescents aged 13 to 17 also found that 20% of adolescents are depressed and that 10% of them have attempted suicide [63]. On the contrary, social connection has a significant positive relation to improved mental wellbeing. The most important predictors of teenage psychosocial health are typically thought to be social and interpersonal relationships [8]. A factor that could contribute to assisting teenagers to cope with stress and can be used to anticipate their mental health is their positive relationships with family and friends. This finding is analogous to studies by Liu and Merritt on Chinese children and adolescents [44]. Therefore, involvement in different groups, such as sports, religious, and heritage groups, along with informal activities with friends improve teenagers’ social connections and assist in the improvement of their metal wellbeing. These findings are in line with the results reported by Badri et al. [8] and Elsina and Martinsone [52].

Our study found that self-perceived depression mediated the relationship between social connection and improved mental wellbeing in a sample of Malaysian teenagers. Hence, teenagers’ improved mental wellbeing could be determined by their social connections if proper guidance and support were provided to reduce self-perceived depression among teenagers by helping to more effectively reduce or handle adolescent depression and stress. These findings support a meta-analysis by Eccles and Qualter [64] on the role of social interactions in improving adolescents’ mental wellbeing. With this in mind, the present study determines that teenagers who would not normally be able to easily obtain formal mental health care can receive these services [62,65]. Since there is no single reason for depression, there is also no single remedy for the condition. Multiple approaches can be helpful for developing the social connectivity of teenagers and allowing them to live in an environment conducive to feeling at ease in expressing their thoughts and ideas; connecting them with the surrounding environment and making them feel valuable and purposeful can be helpful for reducing their anxiety and depression [66].

Social science research ontologically faces constraints. The present study also has some limitations. Since the data were cross-sectional, this limits the causal inference regarding associations among the variables (28). In addition, the study was limited to Malaysian adolescents between the ages of 15 and 19 only, and teenagers from different socio-economic and cultural contexts might behave differently. 

## 7. Conclusions

According to Abaido [7], teenagers’ growing reliance on the Internet and social media has led to reduced social interaction with family members, the deterioration of study, and increased exposure to online games, violence, and bullying, all of which are detrimental to their mental health. Therefore, the purpose of this study was to ascertain how social support and self-perceived sadness affect the mental health of teenagers in Klang Valley, Malaysia. 

Since stronger social support can result in reduced self-perceived depression levels among the Malaysian teenagers addressed in this study, their mental wellbeing can be strengthened to prepare them to move the country forward with meaningful work. Moreover, ensuring healthy lives and promoting wellbeing for all people at all ages is also one of the United Nations’ Sustainable Development Goals [61].

This study has implications for social policies, social services, and social work treatments for teenagers in Malaysia and produces new foundational knowledge that advances our understanding of depression theories. These findings show that stronger social support can result in reduced self-perceived depression among the Malaysian teenagers addressed in this study, and that their mental wellbeing can be strengthened to prepare them to move the country forward with meaningful work. Hence, this study’s findings will have a far-reaching impact towards improving the mental wellbeing of Malaysian teenagers owing to their strong relevancy to government policies that are in line with the Malaysian Strategic Thrusts towards improving wellbeing for all and the UN’s Sustainable Development Goal of good health and wellbeing.

This study recommends that future studies be conducted using samples of all the teenagers in Malaysia.

## Figures and Tables

**Figure 1 ijerph-19-15791-f001:**
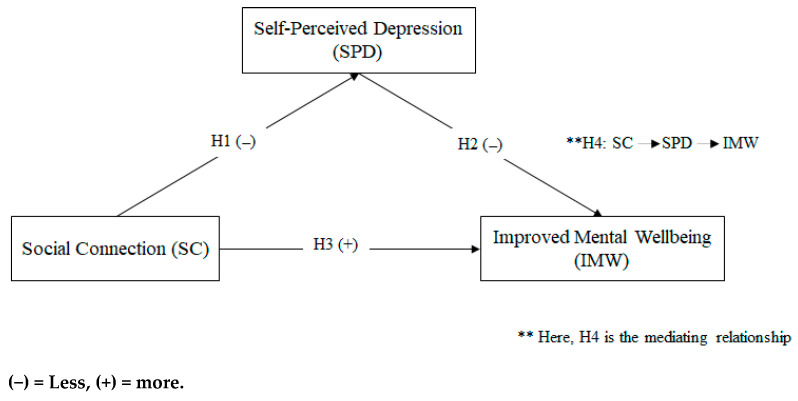
Conceptual framework.

**Figure 2 ijerph-19-15791-f002:**
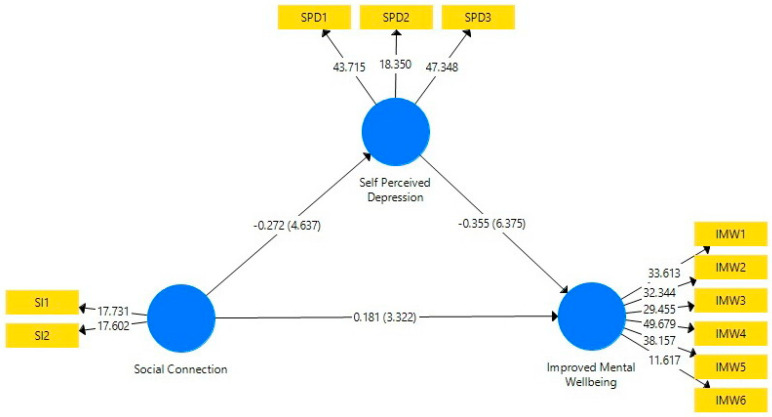
Structural model of the study.

**Table 1 ijerph-19-15791-t001:** Demographic profile of the respondents.

	Frequency	Percent
Gender		
Male	173	59.9
Female	116	40.1
Total	289	100
Age		
Less than 15	87	30
15–19	202	70
Total	289	100
Education		
High school	105	36
Diploma	12	4.2
Bachelor degree	182	59.8
Total	289	100

**Table 2 ijerph-19-15791-t002:** Cronbach’s alpha, composite reliability, and AVE of the measures.

	Cronbach’s Alpha	CR	AVE
Improved mental wellbeing	0.903	0.926	0.678
Self-perceived depression	0.833	0.899	0.748
Social connection	0.643	0.848	0.736

Note: CR: composite reliability, AVE: average variance extracted.

**Table 3 ijerph-19-15791-t003:** Discriminant validity (heterotrait–monotrait ratio (HTMT)).

HTMT	Improved Mental Wellbeing	Self-Perceived Depression
Self-perceived depression	0.457	
Social connection	0.368	0.359

**Table 4 ijerph-19-15791-t004:** R square and Q square.

	R^2^	Q²
Improved mental wellbeing	0.194	0.126
Self-perceived depression	0.074	0.049

**Table 5 ijerph-19-15791-t005:** f square.

f^2^	Improved Mental Wellbeing	Self-Perceived Depression
Self-perceived depression	0.144	
Social connection	0.038	0.080

**Table 6 ijerph-19-15791-t006:** Total effects.

Hypothesis	Original Sample	Sample Mean	Standard Deviation	T Statistics	*p* Values
Social connection -> self-perceived depression (H1)	−0.272	−0.273	0.059	4.637	0.000
Self-perceived depression -> improved mental wellbeing (H2)	−0.355	−0.360	0.053	6.691	0.000
Social connection -> improved mental wellbeing (H3)	0.181	0.187	0.055	3.322	0.000
Mediation Social connection -> self-perceived depression -> improved mental wellbeing (H4)	0.097	0.100	0.027	3.667	0.000

## Data Availability

Data are available upon reasonable request.

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
