# Peer review of "Social Connections and Self-Perceived Depression: An Enhanced Model for Studying Teenagers’ Mental Wellbeing"

_ijerph, 2022, doi:10.3390/ijerph192315791_

Round 1

Author Response

Dear Reviewers,

Thank you for your comments and guidance for improvements.

please find the revised version and response table.

Regards,

MM.

Reviewer 2 Report

Dear authors,

Please find my comments in the document attached.

Author Response

(The authors gave the same response as above.)

Reviewer 3 Report

I would like to thank the authors for their work. You can see that there is a lot of effort for its creation. Below I add information that I hope will serve to improve the manuscript.

Abstract

It is important that the authors add quantitative data and a sentence justifying the need for research on the problem.

Introduction

·         The beginning of the introduction is confusing. The authors reflect all the variables of the paper in the first paragraph. It is recommended to start from the general to the specific by giving international and local data. Therefore, they should start by stating the health problem to be addressed in the study and, little by little, explain the related factors and their relevance to the health problem.

·         It is appropriate for them to use references that are reviews or meta-analyses to support their work. However, claims are made without citations “7.9% of kids…””an extensive theoretical literatura has highlihted…” 

·         The title talks about adolescents but in the introduction and the information on existing literature they present scientific information on children. The processes that predict depression in children, adolescents and adults may be different. Therefore, it is important that literature related to the study population is used and, if information on another population is presented, it has to be cohesive with the text and justified.

·         When they talk about mental wellbeing, they include information from other studies and talk about "mental health". Mental health is not the same as mental well-being as a two-dimensional concept. Maybe it is a mistake in the spelling. Furthermore, they indicate that well-being includes psychological well-being, subjective well-being and complex well-being. However, they do not differentiate between them in the introduction when they point to studies of previous literature and talk about well-being in general. If possible, it would be better to specify which well-being is being referred to.

·         Data on depression is given but no data on well-being is discussed.

·         The importance of the three separate variables and the study population is clear. But there is a lack of information linking the 3 variables and the study population and why this work is important. What information is important to study and why. What is the gap in science about the causes and causes of depression in Malaysian adolescents?

·         A Muslim population is chosen but in the introduction there is no information on the relationship between religion and the study variables.

Methods

·         It is included that questions were asked about variables. However, it is not stated in the document whether information was returned to people about the results of their answers or information about where to seek help if they answered yes to the questions about depressive symptoms. Was any information returned to participants and did they receive anything for answering? This would need to be included in the method.

·         The reliability of the scales should also be indicated, or at least indicate that scales were used or not, adapted to the population and with good psychometric properties. However, this information about the scales should be given even with little information.

·         Information on approval by an ethics committee and consent of guardians or parents is required.

·         It is necessary to know the procedure, how the information was collected, who collected it, etc.

·         Authors can mention the hypotheses when indicating the statistical analysis to be carried out so that it can be seen which analysis is used for each hypothesis.

Results and Discussion

·         Tables have to be self-explanatory. They need to indicate footnotes to the table with the acronyms included.

·         Deben revisar la normativa APA a la hora de poner decimales.

·         The discussion should be separate from the results.

·         The discussion is very poor, lacking information and comparison with other studies.

·         The authors do not include limitations of their study or implications for future lines of research.

Conclusion

How can prevention be carried out to significantly improve depression? It is stated in the conclusion but not in practical terms. They also do not mention whether there are already programmes that include this or whether this should be done from primary care, with clinical adolescents, general population ... They should improve the conclusion to reflect the relevance of the work they present.

Author Response

(The authors gave the same response as above.)

Round 2

Reviewer 1 Report

I would like to commend the authors for their thorough job on revising the manuscript. They have adequately addressed my concerns and its contents have improved significantly.

Author Response

Dear Reviewer,

We have addressed all the comments you have provided in the first round of review process. We have further made some changes and improved the paper. thank you for your supports.

Regards,

MM.

Reviewer 2 Report

Dear authors,

Please find my comments in the document attached.

Kind regards

Author Response

Dear Reviewer,

Thank you very much for your comments and guidance for improvement of our paper. We have made the changes you recommended. Please find the response table and the revised version of the paper.

Thank you,

MM.
